# Epidemiology of Hypoalbuminemia in Hospitalized Patients: A Clinical Matter or an Emerging Public Health Problem?

**DOI:** 10.3390/nu12123656

**Published:** 2020-11-27

**Authors:** Stefania Moramarco, Laura Morciano, Luca Morucci, Mario Messinese, Paola Gualtieri, Mariachiara Carestia, Fausto Ciccacci, Stefano Orlando, Ersilia Buonomo, Jacopo Maria Legramante, Antonino De Lorenzo, Leonardo Palombi

**Affiliations:** 1Section of Hygiene and Public Health, Department of Biomedicine and Prevention, University of Rome Tor Vergata, Via Montpellier, 1-00133 Rome, Italy; laura.morciano@gmail.com (L.M.); lucamorucci88@gmail.com (L.M.); mario.messinese@students.uniroma2.eu (M.M.); mariachiara.carestia@uniroma2.it (M.C.); stefano.orlando@uniroma2.it (S.O.); buonome@uniroma2.it (E.B.); palombi@uniroma2.it (L.P.); 2Section of Clinical Nutrition and Nutrigenomics, Department of Biomedicine and Prevention, University of Rome Tor Vergata, Via Montpellier, 1-00133 Rome, Italy; paola.gualtieri@uniroma2.it (P.G.); delorenzo@uniroma2.it (A.D.L.); 3Unicamillus, International Medical University in Rome, Via di Sant’Alessandro, 8-00131 Rome, Italy; fausto.ciccacci@gmail.com; 4Department of Systems Medicine, University of Rome Tor Vergata, Via Montpellier, 1-00133 Rome, Italy; legramante@med.uniroma2.it

**Keywords:** elderly, fragile populations, hospitalization, hypoalbuminemia, public health, serum albumin

## Abstract

Serum albumin levels are strongly associated with the morbidity, prognosis, and mortality rates of patients with hypoalbuminemia, which is a frequent problem during hospitalization. An observational retrospective study was carried out to analyze changes in albumin levels in hospitalized patients at the “Fondazione Policlinico Tor Vergata—PTV” in 2018. The prevalence of preexisting hypoalbuminemia at the time of discharge from hospital was investigated using a sample of 9428 patients. Information was collected from the discharge files recorded in the central informatics system of the hospital. Analysis of albumin levels at admission and at discharge was conducted by classes of albuminemia and then stratified by age. At the time of admission, hypoalbuminemia was found to be present in more than half of the sample, with no sex differences. The serum albumin level tended to decrease with age, with pathologic levels appearing from 50 years and progressive worsening thereafter. The condition of marked and mild hypoalbuminemia was more prevalent in patients over 65 years of age. Our findings suggest that hypoalbuminemia should be considered a dangerous condition in itself and a serious public health problem. We aimed to emphasize the role of albumin as useful marker of the in-hospital malnutrition and frailty, to be integrated in the routinely assessment of patients for reconsidering ad hoc healthcare pathways after discharge from hospital, especially when dealing with fragile populations.

## 1. Introduction

Human serum albumin is an important parameter for the routine assessment of the nutritional status of patients with acute and chronic conditions [1]. Additionally, it is a recognized valuable biomarker of many diseases, such as cancer [2], ischemia [3], and obesity [4], and is used for monitoring inflammatory activity. Inflammation is a well-known cause of hypoalbuminemia in a number of diseases, including rheumatoid arthritis [5,6]. Albumin is also associated with diseases related to the control of glycemia and adipose tissue [7,8].

In clinical practice, hypoalbuminemia [9] is commonly discovered in association with nutritional deterioration and disease-related inflammatory response [10]. Along with the evolution of the disease itself, this condition might be a result of the aging process, with levels of albumin decreasing with advancing age [11]. However, the relationship between hypoalbuminemia and age has not been fully elucidated; therefore, the association should also take into account diseases and other age-related conditions rather than age alone [12].

Serum albumin levels are strongly associated with morbidity, prognosis, and mortality in both acute and chronic disease patients [13,14], and hypoalbuminemia is a frequent problem in hospitalized patients. Hypoalbuminemia is directly associated with the likelihood of developing frailty conditions [15] and can predict outcome in critically ill patients [16,17] and mortality regardless of comorbidity factors in emergency medical patients [18]. This condition leads to prolonged or recurrent hospitalization, with additional medical costs derived from consequently more expensive treatments for a more efficient management of patients, including the need of extra medical resources. [19,20].

In addition, albumin is used in some prognostic indices, such as the Prognostic Nutritional Index and the Prognostic Inflammatory Nutritional Index. Thanks to this, it is possible to evaluate the relationship of albumin with some solid tumors (colorectal, gastric, pancreas, etc.) and the state of inflammation and malnutrition [21,22].

The present study analyzed changes in albumin levels in hospitalized patients by assessing the prevalence of preexisting hypoalbuminemia at the time of discharge from hospital. The main goal was to support the evidence that low albumin levels still need to be regarded as a dangerous condition in itself and a serious public health problem. By reconsidering the importance of the inclusion of hypoalbuminemia as a specific diagnosis in hospital discharge files, we aimed to propose albumin and its related factors as a reliable biomarker of socio-economic disadvantage for reconsidering ad hoc healthcare pathways for patients after hospital discharge, especially when dealing with fragile populations.

## 2. Materials and Methods

An observational retrospective study was carried out on the entire hospital population of the “Fondazione Policlinico Tor Vergata—PTV” in 2018. Hospital discharge files of admitted patients were collected from the central informatics system of the hospital. Data were extracted using the informatics system AREAS-ADT, with diagnosis coded using the ICD-9 classification (2007 version). All ordinary hospitalizations were included in the analysis (code 1), while all extraordinary hospitalizations (code 2-Day Hospital, code 3-Home treatment, code 4-Day-Surgery with overnight stay) were not considered. Only data of patients above 18 years of age were included in the analysis.

Levels of albumin for the same period of reference were obtained from the informatics system of the Unit of Laboratory Medicine—U.O.C. Medicina di Laboratorio (Modulab)—of the hospital, which stores all the information of samples collected from the admitted patients. Information was collected on the day of the blood sample and a nosology code was assigned to every inpatient event.

A database associating the hospital discharge files with the nosology code was generated. Information included personal data, principal diagnosis, secondary diagnosis (from 1° to 5°), procedures applied during hospitalization, and date on blood sample. In case of multiple admissions, only the first and last values of albumin were analyzed. No anthropometric measurements were available in the database.

To ensure privacy, all data were coded, without personal names.

Consent for data management, analysis, and publication was obtained from the Ethical Committee of the “Fondazione Policlinico Tor Vergata—PTV” (identification number 33/19). The study was conducted in compliance with the Ethical Principles for Medical Research Involving Human subjects of the World Medical Association Declaration of Helsinki (1975).

### Statistical Analysis

Data were analyzed using the Statistical Package for the Social Sciences—SPSS version 24 (IBM, Somers, NY, USA).

A descriptive analysis was performed for the entire study population and subgroups according to age and serum albumin levels.

Four categories of albuminemia were presented in accordance with the classification of previous studies: marked hypoalbuminemia (<2.5 mg/dL), mild hypoalbuminemia (2.5–3.5 mg/dL), normal albumin (3.5–4.5 mg/dL), and hyperalbuminemia (>4.5 mg/dL) [16,23,24,25].

Analysis of albumin levels at admission and at discharge was conducted by dividing the sample into patients below and above 65 years of age. In addition, age was stratified into six classes according to the classification of Akirov et al. [25]: <40, 40–50, 50–60, 60–70, 70–80, and >80 years. The hospitalization diagnoses (ICD-9 coded) were grouped according to the Major Diagnostic Categories (MDC24), then to the Diagnosis Related Groups (DRG24).

Continuous variables were presented as means with standard deviations (SD), while categorical variables were presented as percentages. The statistical significance of differences between albumin levels among groups were assessed with Student’s t-test or ANOVA with Bonferroni post-hoc test for differences among groups, while the chi-squared test was used to compare categorical variables.

Odds ratios and 95% confidence intervals (OR; 95% CI) were calculated to assess the relationship of the risk of mortality with pathologic serum albumin and old age.

Binary logistic regressions were performed to investigate socio-demographic aspects and individual factors that affected different types of outcomes. Because there were multiple independent variables, a stepwise forward regression approach was used.

## 3. Results

A total of 9428 ordinary hospitalizations were recorded in 2018. Of the patients, 55.6% were male. The mean age was 65.2 years ± 16.8 SD, and the median age was 68 years. More than half of the events (57.5%) were recorded for people above 65 years of age. Information on citizenship, marital status, education level, and respective levels of albumin is presented in Table 1. Over 90% of the patients were Italian. As regards marital status, nearly 77% of the participants were married, followed by singles (over 17%). The most frequent educational level among the patients was intermediate school (65.9%), followed by secondary school (16.7%), and primary school (13.2%). A small percentage of patients had the highest educational level (3.4%), while the percentage of illiterate patients was negligible (less than 1%).

Albumin dosage at admission was available for 9367 records, with an overall mean level of 3.389 mg/dL ± 0.634 SD. Albuminemia decreased with increasing age and was below the level of normality (<3.5 mg/dL) in the 50–60 years age group. All the values of albumin were significantly different, with the exception of the 40–50 years age group compared with the 50–60 years age group.

As regards socio-demographic information, albumin levels were statistically lower in Italians (*p* < 0.001), in married and widowed patients compared with single patients (*p* < 0.001 and *p* = 0.035, respectively), and in patients with lower educational levels (primary and intermediate) compared with higher levels (secondary and higher).

The results of the stratification of albuminemia into categories at the time of admission are reported in Table 2. Mean values for all categories were all statistically significant (ANOVA test; *p* < 0.001). Nearly half of the sample already had hypoalbuminemia at their first visit to the hospital (42.9% mild and 9.6% marked), while only slightly less than 45% of patients had a normal level of albumin. The condition of insufficiency albuminemia was significantly more prevalent in elderly patients than in their younger counterparts (67.4% vs. 32.6% for marked; 67.1% vs. 32.9% for mild). Conversely, normal or even highest levels of albumin were more prevalent in patients aged <65 years than in patients aged >65 years (52.2% vs. 47.8% for normal albuminemia; 78.5% vs. 21.5% for hyperalbuminemia; *p* < 0.001) (Figure 1).

The reasons for hospitalization, classified according to the MDC24 categories, are reported in Table 3. The Table shows the numbers and percentages of events recorded by Major Diagnosis Categories. Mean values of albumin by MDC24 at baseline have also been reported. Patients were hospitalized primarily for diseases and disorders of the nervous system (MDC1)—accounting for nearly the twenty percent of all the events, followed by those of the respiratory system (MDC4)—14%-, circulatory system (MDC5)—nearly thirteen percent—those for musculoskeletal system (MDC8)—12.2%, and those of digestive system (MDC6)—slightly more than ten percent. When considering albumin level at baseline, the lowest mean value was found in patients admitted for infectious and parasitic -MCD18 (2.861 mg/dL ± 0.591 SD), while the highest was found in patients admitted for diseases and disorders of the eye-MCD2 (3.991 mg/dL ± 0.402 SD).

In order to better understand the specific diagnosis recorded in the hospital by the period of observation, Table 4 presents the events with a frequency of more than 100 in ascending order, coded by Diagnosis Related Groups (DRG) version 24. Levels of albumin at baseline by DRG24 have also been reported, with significant differences among mean values been investigated. The lowest albumin levels were found for patients admitted because of major cardiothoracic surgeries (MCD5–DRG104; 2.800 mg/dL ± 0.562 SD), while the highest values were found for patients with other factors affecting the state of health (MCD23, DRG467; 3.900mg/dL ± 0.624 SD).

Data corresponding to the second dose of albumin at the time of discharge (6508 cases) were available in the system. The mean albumin level at the second dose was 3.132 mg/dL ± 0.59 SD. Findings on socio-demographic conditions and albumin levels at baseline were also confirmed at the second dose, with lower levels in Italians, as well as in married and widowed patients, compared with single patients. Lower levels of albumin were also observed in patients with lower education (primary and intermediate) compared with those with a higher educational level (secondary and higher), and in illiterate patients compared with those who had attended secondary school (*p* = 0.017).

Albuminemia at discharge stratified by category is reported in Table 5. Mean values for all categories were all statistically significant. Hypoalbuminemia increased in our sample, accounting for more than 70% of the cases (14% marked and 57% mild, respectively). Figure 2 shows that at discharge, hypoalbuminemia was very frequent, with the highest prevalence in elderly patients (*p* < 0.001). Normal levels of albumin, especially hyperalbuminemia, were significantly reduced compared with the measurements at the time of admission (Figure 3).

The mean length of hospital stay was 11.38 ± 13.01 (SD) days, ranging from a minimum of 1 to 223 days.

As regards the outcomes (Figure 4), 47.6% of the admissions were discharged to their homes, 18.4% were transferred to another health facility, while 6.5% died. The remaining 27.5% corresponded to other types of discharges.

Table 6 presents the mean levels of albumin at admission in relation to outcome at discharge from hospital. Patients who died had the lowest level of albumin (2.839 mg/dL ± 0.669 SD), which was significantly lower than that of patients with all other outcomes (*p* < 0.001), including patients transferred to their home (3.222 mg/dL ± 0.598 SD; *p* < 0.001). No difference was found in the levels of albumin between patients discharged home and those with other types of outcomes.

An increased risk of dying was found in patients with pathologic levels of albumin (<3.5 mg/dL) at the time of admission (OR = 4.720; 95% CI: 3.822–5.830). Elderly patients over 65 years of age were at higher risk of dying (OR = 2.468; 95% CI: 2.042–2.983).

Binary logistic regressions were performed to investigate general characteristics (sex, age), socio-demographic factors (marital status, education, citizenship-Italians vs. foreign), and specific conditions (level of albumin at admission) mainly impacting the different types of outcomes (Table 7).

Being divorced or widowed were the two main conditions associated with an increased risk of dying (OR 2.745 and 2.561, respectively). Male patients showed a slightly higher risk than female patients (OR 1.376). With increasing age, there was a tendency for an increase in deaths. Conversely, higher albumin levels at admission were protective factors (OR 0.270; CI 0.236–0.310).

Being single (OR 0.811), having a higher education (OR 0.765), and being young (OR 0.995) were protective factors for being discharged home. The probability of being discharged o home increased with higher levels of albumin at admission.

Being single, female, and older and having a secondary or higher education were all conditions more likely related to the outcome of patients transferred to other health facilities. Conversely, higher albumin levels at admission were protective factors (OR 0.699; CI 0.641–0.762).

Being single and having a higher level of albumin at admission were the main factors associated with an increased risk of other types of discharge. On the other hand, having a secondary school education and being old were protective conditions.

## 4. Discussion

The present study analyzed data of patients admitted to the hospital “Fondazione Policlinico Tor Vergata—PTV” in 2018. Specifically, we investigated and reported the level of serum albumin according to demographic and socio-economic characteristics of patients, admitted for different causes classified as MDC24.

At the time of admission, hypoalbuminemia was found to be present in more than half of our sample, with no sex differences. The socio-demographic analysis revealed that the serum albumin level was significantly lower in patients who were less educated, with illiterate patients or those having primary or intermediate education presenting hypoalbuminemia. Widow and married patients had a significantly lower level of albumin than single and separated/divorced patients.

As regards Major Diagnostic Categories (MDC24), the lowest level of albumin (≤3.0 mg/dL) was reported, as expected, in patients admitted for infectious and parasitic diseases (MDC18), burns (MDC22), multiple significant trauma (MDC24), injuries, and poison and toxic effect of drugs (MDC21), as result of physiological conditions associated with stress, catabolic state, inflammation [9] and hemorrhage. These findings were confirmed when considering the Diagnosis Related Groups (DRG version 24), when the lowest values of albumin (≤3.0 mg/dL) were found, as might be expected, for main surgeries such as major cardiothoracic surgeries (DRG104) and hip and femur surgery (DRG210), as well as for cirrhosis and alcoholic hepatitis (DRG202). More impressive were the results of low albumin levels (nearly 3.1 mg/dL) when simple pneumonia and pleurisy (DGR89) and pulmonary edema and respiratory failure (DRG87) occurred, underlining a possible correlation between infectious diseases and hypoalbuminemia. Future analyses will allow further investigations of the associations and differences between albuminemia, clinical conditions of patients during hospitalization and outcomes for MDC and DRG. These specific findings have not been the purpose of the current paper.

During hospitalization, the serum albumin level tended to decrease with age, with pathologic levels appearing from 50 years, and then progressively worsening. The condition of marked and mild hypoalbuminemia was more prevalent in elderly patients (over 65 years old); conversely, normal and hyperalbuminemia was more prevalent in patients below 65 years of age. Due to the lack of anthropometric parameters routinely assessed and entered in the database, we can only hypothesize that a relevant fraction of our patients could be in a malnutrition status already at time of admission, even though we are not currently able to exclude clinical causes (inflammation, cancer, etc.) of hypoalbuminemia. We acknowledge that is one of main limit of the current study. Further investigations will be necessary to clarify this point. However, albumin is widely used as a marker of nutritional condition [12] and is known to be related to age and health status [26].

Additionally, we hypothesize that albumin could be an efficient and reliable biomarker of socio-economic disadvantage. Being separated/divorced or widowed increased the risk of dying during hospitalization, as well as being illiterate or with a low education level. On the opposite, being single or having a higher education was associated with a higher probability of other favorable outcomes, especially being discharged at home. After the frailty concept [27] evolved in two different directions—the first was the so-called biomedical model or phenotypic and clinical model, that privileged the physical performance [28,29]; the second was the so-called bio-psychosocial model, which privileged the multidomain vulnerability, strictly associated with higher health services demand and unfavorable outcomes [30]—indirect signs of frailty according to both definitions have been seen in our sample. We can conclude that albumin might be a very useful biomarker to depict frailty in the population, constituting a hinge variable between the biomedical model and the multidimensional bio-psycho-social model. Indeed, from our findings, it seems to be associated with both approaches.

Other than albumin levels at admission, our results emphasize the importance of the change in albumin levels before discharge. Hospitalization worsened the patients’ hypoalbuminemia, with more than 70% of the sample having a low level of albumin (either marked or mild) at the time of discharge, with elderly patients accounting for nearly 50% of the sample. A correct evaluation of the nutritional status and personalized nutritional intervention [31] represent important tools for the prognosis and quality of life in hospitalized patients, especially cancer patients. In fact, there are clear mechanisms linking nutritional principles to immune function [32]. Immuno-nutrition treatment improves poor nutritional status or severe malnutrition [33]. The lowering of albumin during the hospital stay could be interpreted not only as a consequence of pathologies and surgical treatments, but also a clear sign of barriers to adequate in-hospital nutrition [34]. Future investigations should be conducted to better understand the specific impact of food access during hospitalizations [35].

Among hospitalized patients, mortality was significantly higher in those with mild and marked hypoalbuminemia, concordant with results from numerous studies reporting hypoalbuminemia as a mortality predictor for different morbidities [25,36,37,38,39]. Logistic regression confirmed older age and lower albuminemia to be risk factors for death outcome [40,41].

In addition, we found that even patients transferred to long-term care health facilities had a pathologic level of albumin. Hypoalbuminemia was statistically associated with an increased risk of mortality and of being transferred to long-term health care facilities.

The catchment of patients at risk of malnutrition at an early stage is paramount to undertake proper nutritional treatment, improving their clinical conditions. Given the worrying albumin values obtained during hospitalization, the need to personalize the nutritional approach in the hospital is emphasized as it is an index of malnutrition and wasting of patients. An enhanced recovery after surgery protocol, which has already been validated for the pre- and post-operative period, is a good example of an optimized approach that aims not only to assess and maintain a good state of nutrition but also to obtain optimal recovery, with a reduction in hospital stay. Furthermore, optimizing this protocol reduces the costs related to hospital management and complications. It has been calculated that for 1000 people with colorectal cancer surgery, the total costs would be reduced by about 3.7 million Euros [42].

## 5. Conclusions

The dosage of serum albumin during hospitalization is a routine practice and an easy-to-access process. From our findings, albumin can be considered as a low-cost marker to stratify patients by risk during hospitalization and as an indicator that, combined with a more exhaustive nutrition evaluation as a best practice, can support decision makers to prescribe nutritional support even after discharging.

Therefore, hypoalbuminemia should be regarded as a dangerous condition in itself to be included as a specific diagnosis in hospital discharge files, especially when a routinely assessment of the malnutrition status is not performed. The need to reconsider ad hoc healthcare pathways for patients after discharge from hospital, especially when fragile populations are involved, warrants further investigations to identify the main procedures associated with albumin levels to lower mortality risk. We emphasize the need to report hypoalbuminemia in a wide range of clinical contexts, from hospital to family doctors and other community facilities, and to consider the relevance of this condition due to its association with hospital admission, intra-hospital mortality, and frailty in a general sense.

The growth in the elderly population demands a transition in healthcare, with a range of modifications in order to maintain the quality of life of the population. In addition to the nutritional status classification, reconsidering albumin as a marker of socio-demographic deprivation during and after hospitalization will support health workers to face this emerging public health problem.

## Figures and Tables

**Figure 1 nutrients-12-03656-f001:**
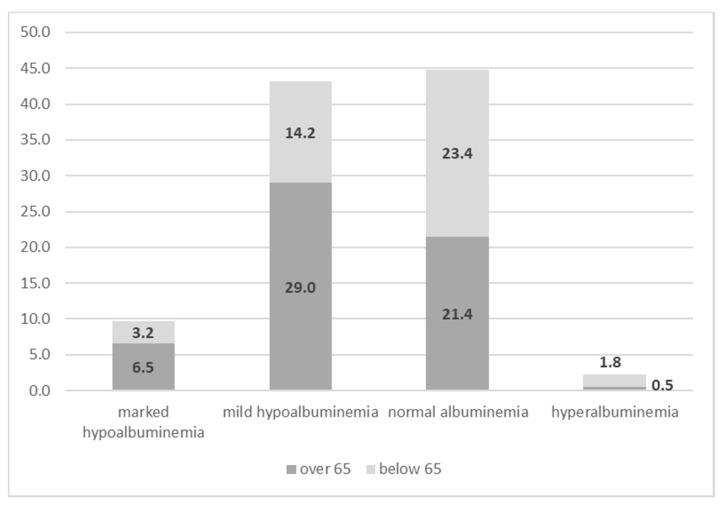
Different levels of albumin at baseline by dichotomized age.

**Figure 2 nutrients-12-03656-f002:**
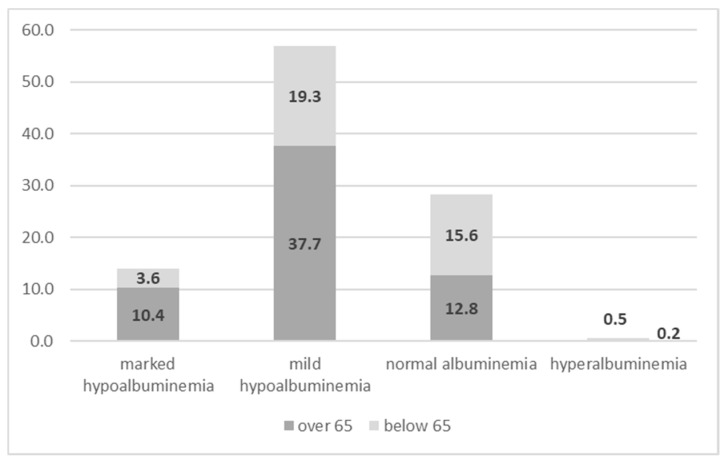
Different levels of albumin at discharge by dichotomized age (below and above 65 years).

**Figure 3 nutrients-12-03656-f003:**
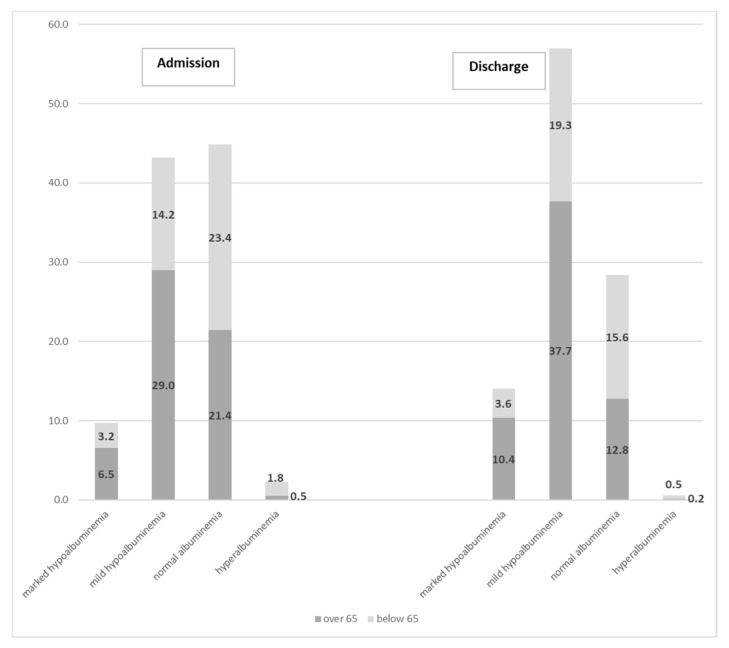
Comparison of prevalence of albumin categories at admission and discharge.

**Figure 4 nutrients-12-03656-f004:**
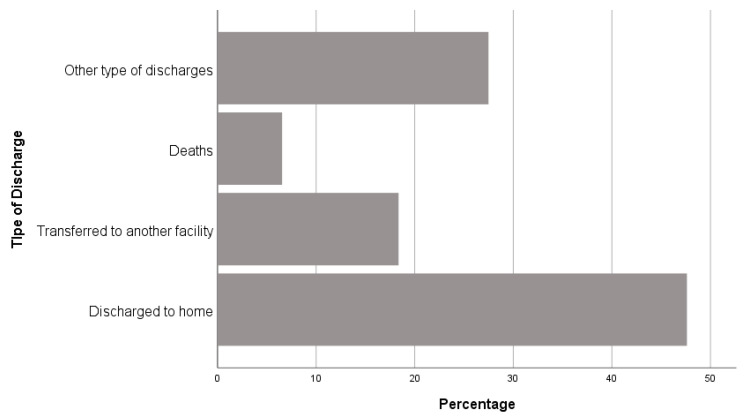
Outcome distribution by categories.

**Table 1 nutrients-12-03656-t001:** General characteristics of the sample at the baseline.

	Variable	Frequency, N. (%)	Albumin Level, Mean (mg/dL) ± SD	*p* Values (ANOVA Test)
**Gender**	Males	5238 (55.6)	3.392 ± 0.683	NS
	Females	4190 (44.4)	3.378 ± 0.614	
**AGE (years)**	<40	886 (9.4)	3.734 ± 0.604	All statistically significant *p* < 0.001 except 40–50 vs. 50–60 NS
	40-50	879 (9.3)	3.558 ± 0.635	
	50-60	1502 (15.9)	3.481 ± 0.641	
	60-70	2067 (21.9)	3.393 ± 0.629	
	70-80	2340 (24.8)	3.306 ± 0.584	
	>80	1754 (18.6)	3.143 ± 0.568	
**CITIZENSHIP**	Italian	8674 (92)	3.376 ± 0.630	
	Foreign	754 (8)	3.501 ± 0.652	<0.001
**MARITAL STATUS**	Single	1638 (17.4)	3.494 ± 0.650	Single vs. Married
	Married	7245 (76.8)	3.360 ± 0.628	<0.001
	Separated/Divorced	218 (2.3)	3.420 ± 0.638	
	Widow	327 (3.5)	3.382 ± 0.597	Single vs. Widow = 0.035
**EDUCATION**	Illiterate	83 (0.9)	3.385 ± 0.593	Prim. vs. Int. <0.001
	Primary School	1242 (13.2)	3.270 ± 0.592	Prim. vs. Sec. <0.001
	Intermediate School	6211 (65.9)	3.357 ± 0.626	Prim. vs. High <0.001
	Secondary School	1570 (16.7)	3.551 ± 0.654	Int. vs. Sec. <0.001
	Higher Education	322 (3.4)	3.578 ± 0.630	Int. vs. High = 0.044

**Table 2 nutrients-12-03656-t002:** Categories of albuminemia at time of admission (baseline).

Albuminemia at Baseline	Frequency n, (%)	Value, Mean Value ± SD	Prevalence in >65 Years Old, n, (%)	Prevalence in <65 Years Old, n, (%)
**Marked hypoalbuminemia** **(<2.5 mg/dL), and**	909 (9.6)	2.177 ± 0.288	613 (67.4)	296 (32.6)
			*p* < 0.001	
**Mild hypoalbuminemia** **(2.5–3.5 mg/dL)**	4045 (42.9)	3.073 ± 0.267	2714 (67.1)	1331 (32.9)
			*p* < 0.001	
**Normal albuminemia** **(3.5–4.5 mg/dL)**	4199 (44.5)	3.884 ± 0.259	2009 (47.8)	2190 (52.2)
			*p* < 0.001	
**Hyperalbuminemia** **(>4.5 mg/dL)**	214 (2.3)	4.645 ± 0.127	46 (21.5)	168 (78.5)
			*p* < 0.001	

**Table 3 nutrients-12-03656-t003:** Hospitalizations by MDC24.

Major Diagnostic Categories (MDC24)	Frequency	Percentage	Albumin at Baseline (mg/dL), Mean ± SD
diseases and disorders of the nervous system **(1)**	1830	19.8	3.663 ± 0.481
diseases and disorders of the respiratory system **(4)**	1299	14.1	3.254 ± 0.621
diseases and disorders of the circulatory system **(5)**	1194	12.9	3.134 ± 0.621
diseases and disorders of the musculoskeletal system and connective tissue **(8)**	1128	12.2	3.471 ± 0.535
diseases and disorders of the digestive system **(6)**	945	10.2	3.324 ± 0.667
diseases and disorders of the hepatobiliary system and pancreas **(7)**	831	9.0	3.291 ± 0.651
myeloproliferative DDs (poorly differentiated neoplasms) **(17)**	429	4.6	3.485 ± 0.616
diseases and disorders of the kidney and urinary tract **(11)**	404	4.4	3.297 ± 0.692
infectious and parasitic DDs (Systemic or unspecified sites) **(18)**	178	1.9	2.861 ± 0.591
factors influencing health status and other contacts with health services **(23)**	177	1.9	3.578 ± 0.674
diseases and disorders of the endocrine, nutritional and metabolic system **(10)**	169	1.8	3.472 ± 0.578
diseases and disorders of the blood and blood forming organs and immunological disorders **(16)**	144	1.6	3.372 ± 0.646
diseases and disorders of the skin, subcutaneous tissue and breast **(9)**	102	1.1	3.328 ± 0.645
diseases and disorders of the ear, nose, mouth and throat **(3)**	96	1.0	3.897 ± 0.56
mental diseases and disorders **(19)**	95	1.0	3.552 ± 0.531
injuries, poison and toxic effect of drugs **(21)**	70	0.8	3.014 ± 0.761
diseases and disorders of the eye **(2)**	51	0.6	3.991 ± 0.402
multiple significant trauma **(24)**	46	0.5	3.027 ± 0.689
human Immunodeficiency virus infection **(25)**	21	0.2	3.235 ±0.845
diseases and disorders of the male reproductive system **(12)**	13	0.1	3.243 ± 0.669
diseases and disorders of the female reproductive system **(13)**	13	0.1	3.328 ± 0.507
alcohol/drug use or induced mental disorders **(20)**	6	0.1	3.418 ± 0.623
newborn and other neonates (perinatal period) **(15)**	3	0	3.483 ± 0.317
burns **(22)**	1	0	3.030
**Total**	9245	100	3.385 ± 0.629

The hospital does not have a department of pregnancy, childbirth, and puerperium (14).

**Table 4 nutrients-12-03656-t004:** Main diagnosis coded by DRG24 of hospitalization.

MDC24	Diagnosis of Hospitalization (DRG24)	Frequencies	Percentages	Albumin at Baseline (Mg/Dl), Mean ± SD	Significant Differences among Drgs: *p* Values <0.05 (ANOVA Test)
1	Transient cerebral ischemia **(524)**	101	1.1	3.670 ± 0.442	vs. 87, 89, 104, 127, 179, 202, 210, 544, 569
17	Acute leukemia without major surgery **(473)**	101	1.1	3.570 ± 0.547	vs. 87, 89, 104, 127, 179, 202, 210, 569
6	Inflammatory diseases of intestine **(179)**	102	1.1	3.300 ± 0.584	vs. 2, 12, 14, 75, 104, 219, 467, 473, 543
6	Major surgery of both large and small intestine **(569)**	104	1.1	3.285 ± 0.826	vs. 2, 12, 14, 75, 219, 410, 467, 473, 524, 543, 544
23	Other factors affecting the state of health **(467)**	110	1.2	3.900 ± 0.624	vs. 87, 89, 104, 127, 179, 202, 210, 410, 544, 569
8	Operation on the lower limb and humerus except hip. foot and femur **(219)**	124	1.3	3.600 ± 0.418	vs. 87, 89, 104, 127, 179, 202, 210, 544, 569
17	Chemotherapy not associated with secondary diagnosis of acute leukemia **(410)**	127	1.3	3.440 ±0.545	vs. 2, 12, 75, 87, 89, 104, 202, 210, 467, 569
1	Degenerative diseases of the nervous system **(12)**	136	1.4	3.830 ± 0.436	vs. 87, 89, 104, 127, 179, 202, 210, 410, 544, 569
1	Craniotomy with major device implant or major diagnosis of complex acute pathology of the central nervous system **(543)**	143	1.5	3.570 ± 0.426	vs. 87, 89, 104, 127, 179, 202, 210, 569
8	Hip and femur surgery. except major joints **(210)**	147	1.6	3.090 ± 0.420	vs. 2, 12, 14, 75, 219, 410, 467, 473, 524, 543, 544
5	Heart valve surgery and other major cardiothoracic surgeries with cardiac catheterization **(104)**	155	1.6	2.800 ± 0.562	vs. 2, 12, 14, 75, 87, 127, 179, 202, 410, 467, 473, 524, 543, 544
1	Craniotomy **(2)**	165	1.8	3.800 ± 0.408	vs. 87, 89, 104, 127, 179, 202, 210, 410, 544, 569
7	Cirrhosis and alcoholic hepatitis **(202)**	177	1.9	3.040 ± 0.665	vs. 2, 12, 14, 75, 104, 219, 410, 467, 473, 524, 543, 544
5	Heart failure and shock **(127)**	179	1.9	3.300 ± 0.516	vs. 2, 12, 14, 75, 87, 104, 219, 467, 473, 524, 543
4	Major interventions on the chest **(75)**	180	1.9	3.770 ± 0.540	vs. 87, 89, 104, 127, 179, 202, 210, 410, 544, 569
4	Pulmonary edema and respiratory failure **(87)**	204	2.2	3.090 ± 0.598	vs. 2, 12, 14, 75, 104, 127, 219, 410, 463, 473, 524, 543, 544
8	Replacement of major joints or reimplantation of the lower limbs **(544)**	305	3.2	3.395 ± 0.449	vs. 2, 12, 14, 75, 87, 89, 104, 202, 210, 219, 467, 524, 569
4	Simple pneumonia and pleurisy **(89)**	325	3.4	3.100 ± 0.569	vs. 2, 12, 14, 75, 127, 219, 410, 467, 473, 524, 543, 544
1	Intracranial hemorrhage or cerebral infarction **(14)**	551	5.8	3.700 ± 0.469	vs. 87, 89, 104, 127, 179, 202, 210, 544, 569
	**Totals**	3,436	36.4	3.403 ± 0.595	

**Table 5 nutrients-12-03656-t005:** Categories of albuminemia at second dosage (discharge).

Albuminemia at Second Dosage	Frequency n, (%)	Value, Mean Value ± SD	Prevalence in Patients over 65 Years Old, n, (%)	Prevalence in Patients below 65 Years Old, n, (%)
**Marked hypoalbuminemia (<2.5 mg/dL), and**	912 (14)	2.156 ± 0.277	677 (74.2)	235 (25.8)
			*p* < 0.001	
**Mild hypoalbuminemia (2.5–3.5 mg/dL)**	3709 (57)	3.019 ± 0.281	2452 (66.1)	1257 (33.9)
			*p* < 0.001	
**Normal albuminemia (3.5–4.5 mg/dL)**	1847 (28.4)	3.808 ± 0.238	830 (44.9)	1017 (55.1)
			*p* < 0.001	
**Hyperalbuminemia (>4.5 mg/dL)**	40 (0.6)	4.678 ± 0.142	10 (25)	30 (75)
			*p* < 0.001	

**Table 6 nutrients-12-03656-t006:** Albumin level by outcomes.

Outcome	Albumin Level, Mean (mg/dL) ± SD	*p* Value (ANOVA Test)
discharged home	3.455 ± 0.610	Death vs. all <0.001 Transferred vs. all <0.001 Discharged to home vs. other type of discharge NS
transferred to long-term health care facilities	3.222 ± 0.598	
death	2.839 ± 0.669	
other type of discharge	3.504 ± 0.597	

**Table 7 nutrients-12-03656-t007:** Impact of different outcomes on general characteristics, socio-demographic factors, and specific conditions.

Outcome	Variable	Exp(B)	Significance	95% CI Per Exp(B)
**Death**	Divorced	2.745	0.029	1.110–6.786
	Widow	2.561	0.004	1.340–4.893
	Sex (male vs. female)	1.376	<0.001	1.153–1.641
	Age	1.029	<0.001	1.023–1.036
	Albumin at admission	0.270	<0.001	0.236–0.310
**Discharged home**	Single	0.811	<0.001	0.724–0.997
	Higher education	0.765	0.020	0.609–0.959
	Age	0.995	<0.001	0.992–0.997
	Albumin at admission	1.328	<0.001	1.241–1.421
**Transferred to long-term health care facilities**	Single	1.213	0.019	1.033–1.424
	Secondary	1.637	<0.001	1.358–1.973
	Higher	1.798	0.002	1.237–2.613
	Sex (male vs. female)	0.821	<0.001	0.738–0.914
	Age	1.018	<0.001	1.014–1.022
	Albumin at admission	0.699	<0.001	0.641–0.762
**Other types**	Single	1.283	<0.001	1.127–1.461
	Secondary	0.825	0.003	0.728–0.935
	Age	0.989	<0.001	0.986–0.992
	Albumin at admission	1.399	<0.001	1.295–1.512

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
