# Peer review of "Epidemiology of Hypoalbuminemia in Hospitalized Patients: A Clinical Matter or an Emerging Public Health Problem?"

_nutrients, 2020, doi:10.3390/nu12123656_

Round 1

Reviewer 1 Report

The article presented to me for review is based on a retrospective study and concerns the analysis of the malnutrition marker, i.e. serum albumin concentration. The finding of a significant percentage of elderly hospitalized patients with lowered albumin concentration is probably nothing new in medicine. However, the proposal to assess this marker in all hospitalized patients is, in my opinion, justified, as is the assessment of nutrition based on anthropometric parameters and with the use of appropriate malnutrition scales. It is necessary to "catch" patients at risk of malnutrition at an early stage, in order to undertake proper nutritional treatment, improving the clinical condition of patients. Unfortunately, the work has no information defining the clinical profile / condition of patients of this study. We do not know with which disease the patients were hospitalized? What methods of nutritional status assessment were used on admission to hospital? Unfortunately, the work concerning only the measurement of serum albumin concentration as a marker in prognosis, seems insufficient with regard to the further clinical fate of patients. The work requires supplementing the data in accordance with the comments of the reviewer.

Author Response

We would like to really thank the advices of the Reviewer#1, that have given us the possibility to improve our paper. 

We are hereby replying point to point to the Reviewer's queries:

Query 1: Unfortunately, the work has no information defining the clinical profile / condition of patients of this study. We do not know with which disease the patients were hospitalized? 

Answer 1: Table 4 has been added in order to better understand the specific diagnosis recorded in the hospital by the period of observation. Table 4 presents the events with a frequency of more than 100 in ascending order, coded by Diagnosis Related Groups-DRG version 24. (see addition at row 101 and from row 156 to 164). Main findings have been discussed from row 248 to 254. We acknowledge the importance for further analyze patients' clinical conditions but these data were not available for our study, since this investigation was beyond the scope of the present paper (lines 254-257).  

Query 2: What methods of nutritional status assessment were used on admission to hospital? Unfortunately, the work concerning only the measurement of serum albumin concentration as a marker in prognosis, seems insufficient with regard to the further clinical fate of patients.   

Aswer 2: No anthropometric measurements were available in the database (row 83). We acknowledged that is one of main limit of the current study (rows 257-260). We recognize that the catchment of patients at risk of malnutrition at an early stage is paramount to undertake proper nutritional treatment, improving their clinical conditions (rows 280-281). Therefore, despite emphasizing that serum albumin concentration combined with a more exhaustive nutrition evaluation should be a best practice (rows 274-278), the present study highlights that albiuminemia, should be regarded as a dangerous condition in itself to be included as a specific diagnosis in hospital discharge files, especially when a routinely assessment of the malnutrition status is not performed (rows 290-292).

Reviewer 2 Report

1. This is an observational retrospective study. It is means that has the disadvantages of this kind of research perse. On the other hand the results are very useful , mainly, from the clinical point of view. 2.The main question addressed by the research is to give attention in patients both in the prevalence of preexisting hypoalbuminemia and also the time of discharge from the hospital. From this point of view the paper is relevant and interesting. 3. Although there are and other papers with similar topic, the fact that we have data from a very large cohort, give us additional knowledge. 4. The text is clear and read to read. 5. The paper is well written. 6 The conclusions consistent with the evidence and arguments presented. 7. Yes, they answer to their hypothesis.

Author Response

We would like to thank to reviewer #2 for his/her positive feedback.

Best regards

This manuscript is a resubmission of an earlier submission. The following is a list of the peer review reports and author responses from that submission.